# Beyond-mean-field analysis of the Townes soliton and its breathing mode

D. S. Petrov[1]

Université Paris-Saclay, CNRS, LPTMS, 91405 Orsay, France

* dmitry.petrov@universite-paris-saclay.fr

April 15, 2025

## Abstract

By using the Bogoliubov perturbation theory we describe the self-bound ground state and excited breathing states of $N$ two-dimensional bosons with zero-range attractive interactions. Our results for the ground state energy $B_N$ and size $R_N$ improve previously known large-$N$ asymptotes and we better understand the crossover to the few-body regime. The oscillatory breathing motion results from the quantum-mechanical breaking of the mean-field scaling symmetry. The breathing-mode frequency scales as $\Omega \propto |B_N|/\sqrt{N}$ at large $N$.

## 1   Introduction

At the mean-field level the problem of two-dimensional attractive bosons is governed by the nonlinear Schrödinger equation with cubic nonlinearity. Its localized stationary solution, called Townes soliton, was found by Chiao and co-workers who studied propagation of optical beams in dielectric materials [1]. Townes solitons have a few peculiar properties related to the scale invariance of the underlying classical mean-field theory [2] (see also [3]).

Namely, the soliton is stationary only when the coupling constant $g$ takes a critical value $g_c$ related to the norm of the wave function $N$ by $g_c N = -\pi C$, where $C = 1.862$. For $g < g_c$ the soliton collapses and for $g > g_c$ it expands. These features have recently been observed in ultra-cold gas experiments [4–6]. Exactly at $g = g_c$ one can generate an infinite number of stationary states by rescaling a unique dimensionless Townes profile by an arbitrary scaling factor. For all these stationary solutions the mean-field energy vanishes [7,8]. In other words, the mean-field theory leaves the size of the soliton undefined and predicts zero for its energy and for its breathing (or monopole) mode frequency.

A different scenario is suggested by various exact results obtained for finite $N$ [9–17]. It is established that the trimer and the tetramer have two (ground and excited) self-bound states with finite energies $B_3 = 16.522688(1)B_2$, $B_3' = 1.2704091(1)B_2$ [13] and $B_4 = 197.3(1)B_2$, $B_4' = 25.5(1)B_2$ [14], respectively. Here, $B_2 < 0$ is the energy of the dimer and the primes denote excited states.

Hammer and Son [13] predicted that the energy and the size of the ground state should scale respectively as $B_N \sim B_2 e^{4N/C}$ and $R_N \sim R_2 e^{-2N/C}$ for large $N$. Their theory is based on the idea that the *renormalized* coupling constant $g$ runs logarithmically with the system size and, therefore, breaks the mean-field scale invariance. Bazak and Petrov [17] have calculated ground-state energies for up to $N = 26$ particles confirming the exponential scaling of Ref. [13]. Moreover, they attempted to fit the results with the ansatz $B_N = B_2 e^{2N/C + c_1 + c_2/N + \cdots}$ arriving at $c_1 \approx -2.06$.

Little is known about excited states for $N > 4$. It is however quite natural to assume (particularly looking at the problem within the hyperspherical formalism [16]) that the $B_N \to B_N'$ excitation is a precursor of the breathing mode. The finite value of $B_N' - B_N$ is thus a beyond-mean-field effect and it emerges as a clear experimentally testable indicator of a quantum anomaly which breaks the mean-field scale symmetry. Olshanii and co-workers [18] proposed to observe this anomaly in a trapped repulsive Bose gas, arguing that the breathing-mode frequency should deviate from two times the trap frequency.

In this paper we develop the Bogoliubov perturbation theory for the two-dimensional soliton. The theory confirms that in the limit of large $N$ the quantity $\ln(B_N/B_2) - 2N/C$ indeed tends to a constant $c_1$ [19]. Our numerical diagonalization of the Bogoliubov-de Gennes equations and calculation of the leading beyond-mean-field correction gives $c_1 = -1.91(1)$. We then discuss the breathing mode of the soliton. Introducing the soliton radius as a collective variable we derive the corresponding equation of motion and obtain for the breathing-mode frequency $\hbar\Omega = 3.804|B_N|/\sqrt{N}$. Our results give quantitative basis for observing quantum effects in droplets with large but finite $N$.

## 2   Mean-field description

The mean-field description of $N$ two-dimensional bosons with contact attraction is obtained from the Lagrangian density

$$\mathcal{L}(\Psi, \Psi^*) = \mathrm{Re}[i\Psi^*(\boldsymbol{\rho}, t)\partial_t \Psi(\boldsymbol{\rho}, t)] - |\nabla_{\boldsymbol{\rho}}\Psi(\boldsymbol{\rho}, t)|^2/2 - g|\Psi(\boldsymbol{\rho}, t)|^4/2, \tag{1}$$

where the coupling constant $g$ is negative, the field $\Psi$ is normalized as $\int d^2\rho |\Psi(\boldsymbol{\rho}, t)|^2 = N$, and we set $\hbar = m = 1$. The equation of motion corresponding to Eq. (1) is the Gross-Pitaevskii equation

$$i\partial_t \Psi(\boldsymbol{\rho}, t) = -(1/2)\nabla_{\boldsymbol{\rho}}^2 \Psi(\boldsymbol{\rho}, t) + g|\Psi(\boldsymbol{\rho}, t)|^2 \Psi(\boldsymbol{\rho}, t), \tag{2}$$

which, for $g = g_c = -\pi C/N$ allows for a family of nodeless stationary solutions

$$\Psi_R(\boldsymbol{\rho}, t) = e^{it/(2R^2)}\Psi_R(\boldsymbol{\rho}) = e^{it/(2R^2)}\sqrt{N/(2\pi C)}f(\rho/R)/R. \tag{3}$$

The dimensionless Townes profile $f$ is a unique nodeless real solution of [1]

$$f''(r) + f'(r)/r + f(r)^3 = f(r). \tag{4}$$

It has a bell-like shape with radius of order one and it satisfies the relations

$$C := \int_0^\infty dr r f^2(r) = \int_0^\infty dr r [f'(r)]^2 = \frac{1}{2} \int_0^\infty dr r f^4(r) = 1.862 \tag{5}$$

and

$$M_2 := \int_0^\infty dr r^3 f^2(r) = 2.211. \tag{6}$$

Using Eqs. (3) and (5) one can show that the mean-field energy functional

$$E_{\mathrm{MF}} = (1/2) \int d^2\rho [|\nabla_{\boldsymbol{\rho}} \Psi(\boldsymbol{\rho}, t)|^2 + g|\Psi(\boldsymbol{\rho}, t)|^4] \tag{7}$$

indeed vanishes independent of the size $R$ when $g = g_c$ and $\Psi = \Psi_R$. In fact, when an arbitrary initial wave function is allowed to evolve according to Eq. (2), the mean square radius (or second moment) of the corresponding density profile evolves according to

$$\partial_t^2 \sigma^2 = 4E_{\mathrm{MF}}/N, \tag{8}$$

where $E_{\mathrm{MF}}$ is the (conserved) mean-field energy given by Eq. (7) [2,3,7,8]. This is why for stationary solutions (3) the energy is necessarily zero. We note however that to extract an atom from the droplet requires energy $-\mu = 1/(2R^2)$. The chemical potential $\mu$ can be deduced either from the explicit time dependence $e^{-i\mu t}$ in Eq. (3) or by calculating the derivative $\mu = \partial_N E_{\mathrm{MF}}$ at fixed $g$ and $R$.

Let us discuss the mean-field breathing dynamics of the Townes soliton, which can be initiated, for instance, by changing $g$ [5]. Consider the ansatz

$$\Psi(\boldsymbol{\rho}, t) = \sqrt{N/(2\pi C)} e^{i\theta(\rho,t)} f[\rho/R(t)]/R(t), \tag{9}$$

which is the Townes profile with time-dependent $R$. The trajectory $R(t)$ and the phase $\theta(\rho, t)$ are subject to variational minimization. We substitute Eq. (9) into the Lagrangian density (1) and minimize the action $S = \int \mathcal{L} d^2\rho dt$ with respect to the field $\theta(\rho, t)$. This gives the continuity equation

$$\partial_t |\Psi|^2 + \nabla_{\boldsymbol{\rho}}(|\Psi|^2 \nabla_{\boldsymbol{\rho}} \theta) = 0, \tag{10}$$

which is solved by

$$\theta(\rho, t) = [\dot{R}(t)/R(t)]\rho^2/2 + \phi(t), \tag{11}$$

where $\phi(t)$ is an arbitrary phase [1]. The action then becomes

$$S_{\mathrm{MF}} = \int dt \left\{ N M_2 \dot{R}^2(t)/(2C) - (g - g_c)N^2/[2\pi C R^2(t)] - N d\phi(t)/dt \right\}. \tag{12}$$

It describes the classical motion of a particle with coordinate $R$ and mass $NM_2/C$ in the potential $(g - g_c)N^2/(2\pi C R^2)$. Note that the kinetic-energy term is the kinetic energy of a superfluid with velocity field $\nabla_{\boldsymbol{\rho}} \theta$ integrated over the soliton profile.

The equation of motion corresponding to the action (12) coincides with Eq. (8), where we replace the mean square radius by its value for the Townes profile $\sigma^2(t) = (M_2/C)R(t)^2$.

---

[1] Since the ansatz (9) forces fixed normalization, the phase $\phi(t)$ just adds a full time derivative to the Lagrangian [see Eq. (12)] and, therefore, plays no role in determining the equation of motion for $R(t)$.

The general solution reads $R(t) = \sqrt{(g - g_c)N/(\pi A) + A(t - t_0)^2/M_2}$, where $A$ and $t_0$ are fixed by the initial conditions. For instance, setting $\dot{R}(0) = 0$ gives the trajectory $R(t) = \sqrt{R^2(0) + (g - g_c)Nt^2/[\pi R^2(0)M_2]}$, which describes expansion for $g > g_c$ or collapse otherwise. When $g = g_c$ the ansatz (9) not only exactly describes the second moment of the soliton but also its full Gross-Pitaevskii evolution, if we set $R(t) = R(0) + Vt$ and $\phi(t) = \int^t dt'/2R^2(t')$ with arbitrary constant $V$.

We point out that for $g \neq g_c$ the true Gross-Pitaevskii evolution leads to corrections beyond the scaling ansatz (9) [2]. In contrast to scaling variations of $\Psi_R$, which conserve the Townes profile, these deviations cost energy and are therefore weak, if the residual term $\int(g - g_c)|\Psi|^4 d^2\rho$ is much smaller than the mean-field kinetic energy or the mean-field interaction energy.

At this point it may be useful to explain the logic behind our perturbative analysis and the hierarchy of energy scales. For stationary Townes solitons described by Eq. (3) (with $g = g_c$) the kinetic and interaction energies equal, respectively, $N/2R^2$ and $-N/2R^2$. We call $N/R^2$ the mean-field energy scale. The quantity $1/R^2$, which is smaller than the mean-field scale by a factor $|g_c| \sim 1/N \ll 1$, will be called the Bogoliubov energy scale. In Sec. 3 we will show that the leading beyond-mean-field correction is of this order of magnitude. If we require $|g - g_c| \sim g_c^2$, the residual mean-field term also belongs to the Bogoliubov level. Corrections at this scale are too weak to significantly change the shape of the soliton, but they govern the dynamics within the degenerate subspace parametrized by $R$. Smallness of $g$ is thus one of the validity conditions of our theory. We will also require that the dynamics of $R$ be sufficiently slow. We will make this more quantitative in the end of Sec. 4. The point is that the kinetic-energy term in Eq. (12) automatically stays at the Bogoliubov level simply due to the energy conservation and it is thus not necessary to know it with higher precision. To promote Eq. (12) to the Bogoliubov accuracy it is sufficient to calculate the leading beyond-mean-field energy correction for a static soliton with $g = g_c$ as a function of $R$.

## 3 Beyond-mean-field analysis

The quantum Hamiltonian directly corresponding to the Lagrangian (1) reads

$$\hat{H} = \frac{1}{2} \int d^2\rho(-\hat{\Psi}_{\boldsymbol{\rho}}^\dagger \nabla_{\boldsymbol{\rho}}^2 \hat{\Psi}_{\boldsymbol{\rho}} + g\hat{\Psi}_{\boldsymbol{\rho}}^\dagger \hat{\Psi}_{\boldsymbol{\rho}}^\dagger \hat{\Psi}_{\boldsymbol{\rho}} \hat{\Psi}_{\boldsymbol{\rho}}), \tag{13}$$

where $\hat{\Psi}_{\boldsymbol{\rho}}^\dagger$ is the operator creating a boson at position $\boldsymbol{\rho}$. The model (13) requires regularization since the two-dimensional delta function $g\delta(\boldsymbol{\rho})$ is not a well-behaved interaction potential (its domain of applicability is limited to mean-field calculations and to the first Born approximation). The regularization can be achieved by introducing a finite interaction range $h$ or by modifying the single-particle dispersion at high momenta $\sim 1/h$. The regularized model approximates the true zero-range model if (a) the range $h$ is much smaller than the typical length scale in the problem (in our case, soliton size $R_N$) and (b) the pair $g$ and $h$ is chosen such that the regularized model reproduces the dimer energy $B_2$, which is a unique parameter characterizing the zero-range interaction.

To calculate the beyond-mean-field term in the Bogoliubov approximation we put the system on a square lattice with spacing $h$ such that $\boldsymbol{\rho}$ runs over the nodes denoted by

---

[2]A simple illustrative example is a quench of a Townes soliton from $g = g_c$ to $g = 0$. After a long free expansion time the profile of the soliton is determined by the initial momentum distribution, but the Fourier transform of $f$ and $f$ itself are of different shapes. Therefore, Eq. (9) cannot be exact in this case.

integers $i$ and $j$. The Laplacian $\nabla_{\boldsymbol{\rho}}^2$ in Eq. (13) is replaced by the lattice Laplacian defined by

$$\hat{L}_h \Phi_{i,j} = (\Phi_{i+1,j} + \Phi_{i-1,j} + \Phi_{i,j+1} + \Phi_{i,j-1} - 4\Phi_{i,j})/h^2, \tag{14}$$

and the integral $\int d^2\rho$ is replaced by $h^2 \sum_{i,j}$. The operator $\hat{L}_h$ is equivalent to $\nabla_{\boldsymbol{\rho}}^2$ at low momenta $p \ll 1/h$. The relation among $B_2$, $g$, and $h$ in this scheme is established by solving the lattice dimer problem. For vanishing center-of-mass momentum we have [20]

$$\frac{1}{g} = -\frac{1}{2\pi(1+|B_2|h^2/4)}\mathbf{K}\left[\frac{1}{1+|B_2|h^2/4}\right] \approx \frac{\ln(|B_2|h^2/32)}{4\pi} - \frac{|B_2|h^2}{32\pi}\ln(|B_2|h^2 e/32), \tag{15}$$

where $\mathbf{K}(\kappa) = \int_0^{\pi/2} d\phi/\sqrt{1 - \kappa^2\sin^2\phi}$ is the complete elliptic integral. The first equality in Eq. (15) is exact and the last expression contains the usual logarithmically running term plus the leading effective-range correction in the limit $|B_2|h^2 \to 0$. We show this expansion to illustrate the difference between two types of terms which typically arise when we study the regularized lattice model. First, there are terms logarithmic in $h$. They are proportional to powers of the small parameter $|g|$ and they are therefore relevant for our perturbative expansion (in powers of $g$). Second, there are terms containing powers of $h \propto e^{2\pi/g}$. Because of their nonanalytic dependence on $g$ they can be neglected at any order. The last term in Eq. (15) is one example of this behavior. Another example is the difference between the lattice (finite $h$) and continuum ($h = 0$) results for the mean-field energy (7). We can also add to this list the discrepancy between the Townes profiles calculated in the continuum and on the lattice. Effective-range corrections are neglected in this manuscript. However, for practical calculations it is useful to know that they exist and behave as $h^2 \ln h$ at small finite $h$ (see Appendix A).

The regularized model allows for a perturbative treatment of the problem of $N$ atoms. Mora and Castin [21] have shown that for the repulsive Bose gas this series goes in integer powers of $g \ll 1$. According to our terminology these are the mean-field term, the Bogoliubov term, the leading beyond-Bogoliubov term, etc. A peculiarity of this hierarchy is related to the freedom of choosing $g$ and $h$ along the curve of constant $B_2$. Varying $g$ we change the mean-field term, but the corresponding variation of $h$ leads to a compensating correction at the Bogoliubov level such that the total result is independent of the choice of $g$ (up to the Bogoliubov accuracy). The bare coupling constant $g$ is thus an auxiliary quantity that we can choose at our convenience (in a certain range) to facilitate the perturbative analysis.

A technical difficulty with the method is that the natural choice $g = g_c = -\pi C/N$ corresponds to $h$ of order the soliton size $R_N$. However, in Ref. [22] we show that for $N \gg 1$ one can choose a significantly smaller $h \ll R_N$, at the same time having $|g - g_c| \sim g^2 \ll |g_c|$. This allows us to formally set $g = g_c$ when calculating the beyond-mean-field correction to the soliton energy.

We now proceed to the standard Bogoliubov theory generalized to the inhomogeneous case [23–26]. The approach consists of writing $\hat{\Psi}_{\boldsymbol{\rho}} = \Psi_R(\boldsymbol{\rho}) + \delta\hat{\Psi}_{\boldsymbol{\rho}}$ and expanding (13) up to second-order terms in powers of $\delta\hat{\Psi}$ and $\delta\hat{\Psi}^\dagger$. The zero-order term is the mean-field energy functional (7) and the first-order terms are absent since we have chosen the condensate wave function satisfying the Gross-Pitaevskii equation. The quadratic terms lead to the Bogoliubov Hamiltonian

$$\hat{H}_2 = \frac{1}{2}\int d^2\rho \left(\delta\hat{\Psi}_{\boldsymbol{\rho}}^\dagger \quad \delta\hat{\Psi}_{\boldsymbol{\rho}}\right)\begin{pmatrix}\hat{A} & \hat{B} \\ \hat{B} & \hat{A}\end{pmatrix}\begin{pmatrix}\delta\hat{\Psi}_{\boldsymbol{\rho}} \\ \delta\hat{\Psi}_{\boldsymbol{\rho}}^\dagger\end{pmatrix} - \text{Tr}(\hat{A})/2, \tag{16}$$

where $\hat{A} = -\hat{L}_h/2 - \mu + 2g_c\Psi_R^2(\boldsymbol{\rho})$, $\hat{B} = g_c\Psi_R^2(\boldsymbol{\rho})$, and we use the fact that $\Psi_R(\boldsymbol{\rho})$ is real. General properties of quadratic Hamiltonians of the type (16) and details of their

diagonalization procedure can be found in the textbook of Blaizot and Ripka [24]. The trace term in Eq. (16) is a consequence of applying the formula $2\delta\hat{\Psi}_{\boldsymbol{\rho}}^{\dagger}\delta\hat{\Psi}_{\boldsymbol{\rho}'} = \delta\hat{\Psi}_{\boldsymbol{\rho}}^{\dagger}\delta\hat{\Psi}_{\boldsymbol{\rho}'} + \delta\hat{\Psi}_{\boldsymbol{\rho}'}\delta\hat{\Psi}_{\boldsymbol{\rho}}^{\dagger} - \delta(\boldsymbol{\rho} - \boldsymbol{\rho}')$ to rearrange creation and annihilation operators in the symmetric manner. The point is that such symmetrized forms stay symmetrized under canonical Bogoliubov transformations. In our case the matrix term in Eq. (16) can be diagonalized into a sum of decoupled harmonic oscillators $\epsilon_{\nu}(\hat{b}_{\nu}^{\dagger}\hat{b}_{\nu} + \hat{b}_{\nu}\hat{b}_{\nu}^{\dagger})/2$ with positive $\epsilon_{\nu}$ plus four "nondiagonalizable" terms of the type $\hat{b}^{\dagger}\hat{b} + \hat{b}\hat{b}^{\dagger} + \hat{b}^{\dagger}\hat{b}^{\dagger} + \hat{b}\hat{b}$ associated to zero modes. The ground state energy of $\hat{H}_2$ equals

$$E_{\mathrm{BMF}} = \sum_{\nu} \epsilon_{\nu}/2 - \mathrm{Tr}(\hat{A})/2. \tag{17}$$

The energies $\epsilon_{\nu}$ are obtained by solving the Bogoliubov-de Gennes equations

$$\begin{pmatrix} \hat{A} & \hat{B} \\ -\hat{B} & -\hat{A} \end{pmatrix} \begin{pmatrix} u_{\nu}(\boldsymbol{\rho}) \\ v_{\nu}(\boldsymbol{\rho}) \end{pmatrix} = \epsilon_{\nu} \begin{pmatrix} u_{\nu}(\boldsymbol{\rho}) \\ v_{\nu}(\boldsymbol{\rho}) \end{pmatrix}. \tag{18}$$

In our case the spectrum of Eq. (18) is real and for each eigenstate $\nu$ with $\epsilon_{\nu} > 0$ there is an eigenstate $\eta$ with $\epsilon_{\eta} = -\epsilon_{\nu}$. This pair of eigenstates corresponds to a physically unique Bogoliubov excitation governed by the Hamiltonian $\epsilon_{\nu}(\hat{b}_{\nu}^{\dagger}\hat{b}_{\nu} + \hat{b}_{\nu}\hat{b}_{\nu}^{\dagger})/2$. The sum in Eq. (17) in our case runs over positive $\epsilon_{\nu}$. Importantly, these modes with finite $\epsilon_{\nu}$ form a scattering continuum above the particle emission threshold, i.e., they are separated from zero by the gap $-\mu = 1/2R^2$.

In addition to these nonzero modes, the Hamiltonian (16) features four zero modes decoupled from one another and from the continuum excitations (we remind that we neglect finite-range effects). These modes are related to arbitrary changes of the complex phase of the condensate wave function [$U(1)$ symmetry], arbitrary shifts of the system in two spatial directions (translational symmetry), and arbitrary changes of $R$ (Pitaevskii-Rosch scale symmetry). In the spectral decomposition of the Hamiltonian (16) the corresponding subspaces can be represented as nilpotent $2 \times 2$ blocks with $\hat{A} = \hat{B} = 1$ or, equivalently, as harmonic oscillators $\hat{p}^2/2 + \omega^2\hat{x}^2/2$ with vanishing frequency $\omega$. These modes describe motion without restoring force. For instance, for the phase mode the "coordinate" $x$ is proportional to the condensate phase and the "momentum" $p$ to the atom number. The main conceptual problem here is that the classical symmetry-broken state (say, localized at coordinate $x = 0$ and momentum $p = 0$) is very different from the quantum-mechanical ground $p = 0$ state, completely delocalized in space. The problem has been discussed in detail in Ref. [24] (see also [25, 26]). Starting with the classical $x = p = 0$ ground state and evolving it with the Hamiltonian $\hat{p}^2/2$ one observes diffusion of $x$. Note, however, that although the linearized theory is formally restricted to small fluctuations around the classical ground state, it does predict the correct (vanishing) energy of the true delocalized state. In other words, we can still use Eq. (17) to predict the energy of the soliton. As long as we are not interested in diffusion effects, the fact that $\Psi_R(\boldsymbol{\rho})$ breaks the symmetries does not prevent from determining the energy. We will return to this point in Sec. 5.

We now discuss the dependence of $E_{\mathrm{BMF}}$ on $h$ and $R$. By using the definitions of the operators $\hat{L}_h$, $\hat{A}$, $\hat{B}$ and the scaling properties of $\Psi_R$ one can check that, if $h$ and $R$ are varied proportionally to each other, the eigenenergy $\epsilon_{\nu}$ scales as $1/R^2$. The quantity $E_{\mathrm{BMF}}$ is thus a product of $1/R^2$ and a function of the ratio $h/R$. Since the operator $\hat{L}_h$ is equivalent to $\nabla_{\boldsymbol{\rho}}^2$ at small momenta the states with $\epsilon_{\nu} \ll 1/h^2$ are insensitive to variations of $h$. Therefore, in the asymptotic limit $h \to 0$ the beyond-mean-field correction Eq. (17) is a constant plus an $h$-dependent part dominated by excitations with energies $\epsilon_{\nu} \gtrsim 1/h^2$. For these high-momentum excitations the matrix in Eq. (18) is dominated by the operator $\hat{L}_h$ inside $\hat{A}$, whereas $\hat{B}$ and all other terms in $\hat{A}$ can be considered as perturbations.

At the zeroth order we deal with eigenfunctions of $\hat{L}_h$, which are plane waves labeled by the two-dimensional momentum vector $\nu = \boldsymbol{p} = \{p_x, p_y\}$ contained in the square $p_x, p_y \in [-\pi/h, \pi/h]$. More precisely, the unperturbed wave functions are $u_{\boldsymbol{p}}^{(0)} = e^{i\boldsymbol{p}\boldsymbol{\rho}}$, $v_{\boldsymbol{p}}^{(0)} = 0$ for positive $\epsilon_{\boldsymbol{p}}^{(0)} = [2 - \cos(p_x h) - \cos(p_y h)]/h^2$ and $u_{\boldsymbol{p}}^{(0)} = 0$, $v_{\boldsymbol{p}}^{(0)} = e^{i\boldsymbol{p}\boldsymbol{\rho}}$ for $-\epsilon_{\boldsymbol{p}}^{(0)}$. One can see that corrections to these eigenvalues coming only from the operator $\hat{A}$ get cancelled by the trace term in Eq. (17). Therefore, the leading nonvanishing contribution in Eq. (17) is related to the second-order correction to $\epsilon_{\boldsymbol{p}}$ induced by the operators $\hat{B}$. The corresponding shift of the eigenvalue equals

$$\delta\epsilon_{\boldsymbol{p}} = -g_c^2 \int \frac{d^2 p'}{(2\pi)^2} \frac{\int d^2\rho d^2\rho' e^{i(\mathbf{p}-\mathbf{p}')(\boldsymbol{\rho}-\boldsymbol{\rho}')}\Psi_R^2(\boldsymbol{\rho})\Psi_R^2(\boldsymbol{\rho}')}{\epsilon_{\boldsymbol{p}}^{(0)} + \epsilon_{\boldsymbol{p}'}^{(0)}} \approx -g_c^2 \frac{\int d^2\rho \Psi_R^4(\boldsymbol{\rho})}{2\epsilon_{\boldsymbol{p}}^{(0)}} = -\frac{N^2 g_c^2}{2\pi C R^2 \epsilon_{\boldsymbol{p}}^{(0)}}.$$
(19)

To derive the second (approximate) equality in Eq. (19) we note that integration over $p'$ of the expression $e^{i(\mathbf{p}-\mathbf{p}')(\boldsymbol{\rho}-\boldsymbol{\rho}')})/(\epsilon_{\boldsymbol{p}}^{(0)} + \epsilon_{\boldsymbol{p}'}^{(0)})$ leads to a kernel which decays exponentially on the scale $|\boldsymbol{\rho} - \boldsymbol{\rho}'| \sim 1/p$. Therefore, for momenta $p \gg 1/R$ we can neglect spatial variations of $\Psi_R$ and assume $\Psi_R(\boldsymbol{\rho}') = \Psi_R(\boldsymbol{\rho})$. The last equality in Eq. (19) follows from Eqs. (3) and (5).

One can now see that the high-energy part of the sum in Eq. (17) is determined by the logarithmic behavior of the integral $\int d^2 p/\epsilon_{\boldsymbol{p}}^{(0)} = \text{const} + 4\pi \ln(R/h)$. Accordingly, for $h \ll R$ we write the beyond-mean-field correction as

$$E_{\mathrm{BMF},h} = -\frac{N^2 g_c^2}{4\pi^2 C R^2} \ln(\xi R/h) = -\frac{C}{4R^2} \ln(\xi R/h),$$
(20)

where we absorb the constant term into the dimensionless parameter $\xi$. We do expect the logarithmic dependence of $E_{\mathrm{BMF},h}$ on $h$ as it has to get compensated by the logarithmic running of the mean-field interaction energy term $(g/2)\int |\Psi_R(\rho)|^4 d^2\rho$ according to Eq. (15). By contrast, the parameter $\xi$ is a valuable quantitative characterization of the beyond-mean-field energy. We point out that $\xi$, defined by Eq. (20) through the asymptotic behavior of $E_{\mathrm{BMF},h}$, is specific to the chosen regularization scheme. On a different lattice or using another single-particle dispersion (with the same low-energy parameters: mass and $B_2$) one would arrive at a different $\xi$. However, similar to the just discussed $h$-invariance, this change would be compensated by a change in Eq. (15). By using the same arguments that led us to Eq. (20) one can check that this cancellation is always automatic as the running of $g$ and change of the beyond-mean-field term both reflect the same change in the high-momentum behavior of the second-order Born integral $\int |U(\mathbf{p})|^2/\epsilon^{(0)}(\mathbf{p})d^2 p$ (here $U$ is the interaction potential in momentum space). One can introduce a truly universal parameter, which depends neither on the regularization scheme nor on $h$. For instance, the combination $4\pi/g - \ln(|B_2|h^2/\xi^2)$ is universal. Let us introduce

$$c_1 = 4\pi/g - \ln(|B_2|h^2/\xi^2) + \ln(C/8) - 1 = -1.91(1),$$
(21)

where the first equality is the definition of $c_1$ and the last number is the result of our numerical analysis discussed in detail in Appendix A. In short, to obtain this number we numerically solve Eq. (18) for $\epsilon_\nu$, then we extract $\xi$ by fitting Eq. (17) with the asymptotic formula (20) at small $h/R$, and finally we use the relation $c_1 = -1 - 8\ln 2 + \ln(C\xi^2)$, which follows from Eqs. (15) and (21). We will now show that $c_1$ defined by Eq. (21) is the large-$N$ limit of $\ln(B_N/B_2) - 4N/C$.

Equation (20) confirms the statement made in the end of Sec. 2 that the Bogoliubov correction is by a factor $|g| \ll 1$ smaller than the mean-field energy. With the Bogoliubov

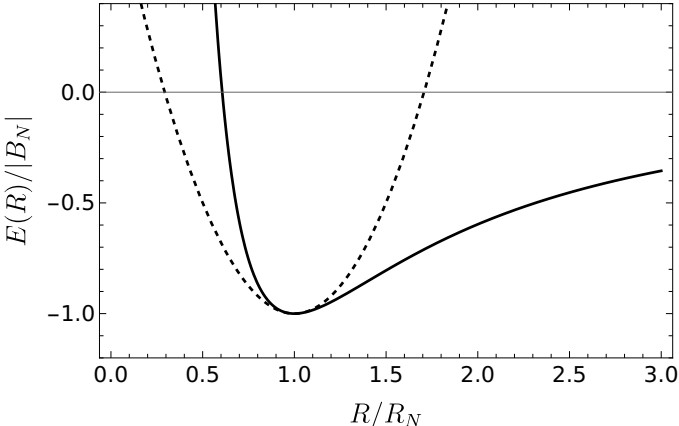

Figure 1: The potential $E(R)$ (solid) describing the breathing dynamics. The dashed curve shows the harmonic approximation valid near the equilibrium point.

accuracy we can now write the soliton energy as

$$E(R) = \frac{(g - g_c)N^2}{2\pi C R^2} - \frac{C}{4R^2} \ln(\xi R/h) = -\frac{N}{2R^2} - \frac{C}{4R^2} \ln \frac{\xi \sqrt{|B_2|} R}{4\sqrt{2}}, \qquad (22)$$

where we use $g - g_c = g g_c(1/g_c - 1/g) \approx g_c^2(1/g_c - 1/g)$ and Eq. (15) neglecting the effective-range term. As expected, $h$ and $g$ drop out of the problem and the result is expressed in terms of the dimer energy $B_2$. Equation (22) has a minimum at

$$R_N = \frac{1}{\sqrt{|B_2|}} e^{-2N/C + 1/2 + \ln(4\sqrt{2}/\xi)} = \sqrt{\frac{C}{8|B_2|}} e^{-2N/C - c_1/2} \qquad (23)$$

with the energy

$$B_N = E(R_N) = -\frac{C}{8R_N^2} = B_2 e^{4N/C - 1 - 8\ln 2 + \ln(C\xi^2)} = B_2 e^{4N/C + c_1 + o(1)}. \qquad (24)$$

The main exponential dependence on $N$ in Eq. (24) has been predicted by Hammer and Son [13]. Here we derive the next-order correction and establish that the quantity $\ln(B_N/B_2) - 4N/C$ tends to $c_1$ given by Eq. (21). One should keep in mind that all our results in this section and in Sec. 4 are valid at the Bogoliubov level. The beyond-Bogoliubov corrections can be presented as the term $o(1)$ shown explicitly in the last exponent in Eq. (24). For brevity we omit it everywhere else.

## 4 Breathing dynamics

Breathing excitations of the Townes soliton are oscillations of $R(t)$ around the equilibrium position given by Eq. (23). To describe them we will use the action (12) in which the potential is replaced by Eq. (22). Omitting the constant term in the action and using Eqs. (23) and (24) we obtain

$$S = \int dt \left[ \frac{NM_2}{C} \frac{\dot{R}^2(t)}{2} - E[R(t)] \right] = \int dt \left[ \frac{NM_2}{C} \frac{\dot{R}^2(t)}{2} + 2|B_N| \frac{R_N^2}{R^2(t)} \ln \frac{R(t)e^{1/2}}{R_N} \right] \quad (25)$$

In Fig. 1 we show the potential $E(R)$. In contrast to the dynamics discussed in Sec. 2, we now deal with a finite oscillating breathing motion. The dashed curve in Fig. 1 shows the

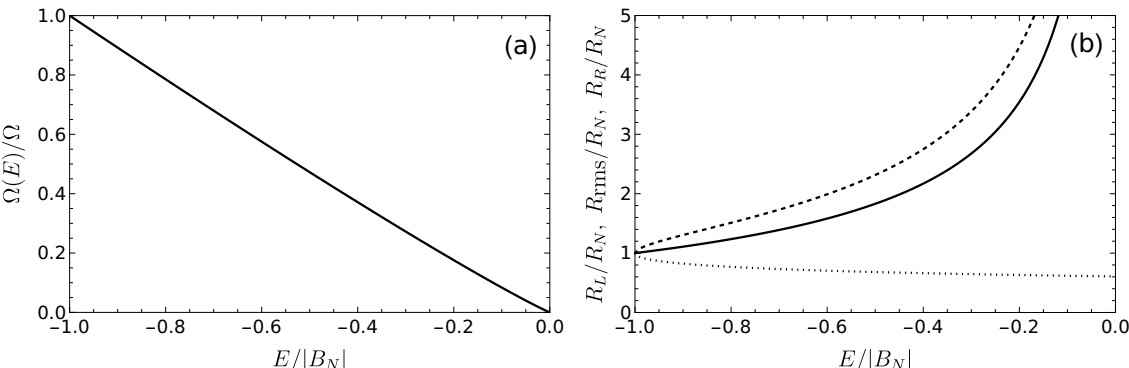

Figure 2: (a) The frequency $\Omega(E)$ of finite-amplitude breathing oscillations as a function of the total energy $E$ of the system in units of $|B_N|$. The limit $E \to -|B_N|$ corresponds to vanishing amplitude, where $\Omega(E) = \Omega$. At the threshold $E = 0$ the soliton has just enough energy to expand to $R \to \infty$. (b) Other parameters of finite-amplitude oscillations: the left and right turning points $R_L$ (dotted) and $R_R$ (dashed) and the rms radius $R_{\rm rms}$ (solid), obtained by averaging $R^2(t)$ over one oscillation period.

harmonic approximation $E(R) \approx -|B_N| + 2|B_N|(R - R_N)^2/R_N^2$ valid in the vicinity of the equilibrium radius. Small-amplitude oscillations are characterized by the frequency

$$\Omega = \frac{4\sqrt{2}}{\sqrt{NM_2}}|B_N|, \tag{26}$$

where $M_2$ is defined in Eq. (6). On the other hand, as one can see from Fig. 1, the harmonic approximation breaks down already for relatively small amplitudes and the oscillation period depends on the excitation amplitude.

Figure 2 illustrates some properties of the breathing dynamics. In Fig. 2(a) we show the finite-amplitude oscillation frequency $\Omega(E)$ in units of $\Omega$ as a function of the conserved total energy of the system $E = NM_2\dot{R}^2/2C + E(R)$ in units of $|B_N|$. Small-amplitude oscillations correspond to $E \approx -|B_N|$. The amplitude increases and the frequency decreases with $E$. When $E$ approaches zero the oscillation period diverges and the droplet spends lots of time at large $R$. In Fig. 2b we show the left turning radius $R_L$ (dotted), the root mean square radius $R_{\rm rms}$ (solid), and the right turning radius $R_R$ (dashed) as a function of $E$. $R_{\rm rms}$ is obtained by averaging $R^2(t)$ over the oscillation period and then taking square root.

To give an idea of realistic time and length scales let us write the full period of small-amplitude oscillations in the dimensional form $\tau = 2\pi/\Omega = 7.1mR_N^2\sqrt{N}/\hbar$. Applying this formula to Cs (we have in mind the quasi-two-dimensional experiments of Chen and Hung [4, 6]) we obtain $\tau \approx 100$ms for $N = 16$ and $R_N = 1.3\mu$m. The corresponding peak density is approximately 4 atoms per $\mu$m$^2$.

We now return to the question of validity of the model (25) and its limitations. In deriving this model we essentially use the Born-Oppenheimer approximation for separating the "fast" Bogoliubov vacuum and the "slow" breathing motion. The idea of adiabatic elimination of fast degrees of freedom to obtain an effective action for slow ones has been proposed to describe molecular dynamics [27] and is widely used in other domains (see, for example, [28–30]). We remind that the nonzero Bogoliubov modes are separated from zero by the gap $1/(2R^2)$. Therefore, according to the adiabatic theorem [31], the Bogoliubov vacuum adiabatically follows the breathing motion, if the rate of change of $R(t)$ is smaller

than the gap. This gives the inequality

$$|\dot{R}(t)/R(t)| \ll 1/R^2(t), \tag{27}$$

which justifies our instantaneous approximation for calculating the beyond-mean-field correction. Note that when doing this calculation in Sec. 3 we use the stationary Townes profile (3) instead of the instantaneous condensate wave function (9). They differ from each other by the $\rho$-dependent phase factor $e^\theta$. However, one can see from Eq. (11) that under the condition (27) spatial variations of $\theta$ over the soliton size remain small and can be neglected.

One can check that for any fixed total energy $E/|B_N| < 0$ and for sufficiently large $N$ the bound periodic trajectories $R(t)$, which minimize the action (25), also satisfy the adiabaticity condition (27). However, at fixed $N$ this condition fails for sufficiently high $E$. In fact, for $E > 0$, the trajectory $R(t)$ goes to infinity. There the potential energy $E(R)$ vanishes, but the kinetic energy remains finite and eventually becomes much larger than the Bogoliubov energy scale, indicating that our approach breaks down. Even for negative $E$, if we take $E > B_{N-1}$, the oscillating motion can be converted into a particle emission. This phenomenon is associated with significant deviations from the ansatz (9) and is not described by the model (25). This is to say that the model (25) does not in general require $|R(t) - R_N| \ll R_N$, but care should be taken when considering large-amplitude oscillations (see also Sec. 5).

## 5 Discussion

Although the mechanism leading to these finite-frequency oscillations is quantum, the dynamics discussed so far in the limit $N \to \infty$ is classical. Let us try to go one step further and imagine what happens when we quantize the classical model (25). First of all, it is clear that in this case the breathing excitations form a discrete ladder. The number of states in this ladder increases with $N$ since the ratio of the level spacing to the well depth $\sim \Omega(E)/|B_N|$ decreases as $1/\sqrt{N}$. Moreover, because of the thick tail of the potential $E(R) \propto -R^{-2} \ln(R/R_N)$ one may even think that there is an infinite number of bound states accumulated just under the threshold $E = 0$. This argument cannot be trusted in view of the fact that the adiabaticity condition (27) gets violated for $R \gtrsim R_N e^{\sqrt{N}}$. This estimate comes from the result $\dot{R}/R \propto R^{-2} N^{-1/2} \ln R/R_N$ obtained for $E = 0$ by equating the kinetic energy $\propto N\dot{R}^2$ and $-E(R)$. Note that for $N = 3$ and $N = 4$ there is only one excited state and an abrupt change from finite number of states to infinite number of states at a certain $N$ seems unrealistic. We thus conjecture that there is a sequence of critical $N$ at which the number of bound states increases by one. We point out that the states are not equidistant, and the quantum breathing dynamics may be quite different from the classical one governed by the periodic trajectory $R(t)$.

Quantization of the breathing mode leads to another interesting observation. Bazak and Petrov [17] have calculated $B_N$ for $N \le 26$ and tried to fit their data with the series expansion $\ln(B_N/B_2) - 4N/C = c_1 + c_2/N + c_3/N^2 + ....$ This expansion is a natural generalization of the perturbative expansion in integer powers of $g$ (in our case $|g| \sim 1/N$) discussed in the repulsive case [21]. We now see that this conjecture is likely to be wrong for the attractive case as the zero-point energy of the breathing mode $\Omega/2$ scales as $1/\sqrt{N}$ suggesting the presence of half-integer powers of $1/N$ in the series. Just adding $\Omega/2$ given by Eq. (26) to Eq. (24) leads to

$$B_N \approx B_2 e^{4N/C + c_1 - 2\sqrt{2}/\sqrt{M_2 N}}, \tag{28}$$

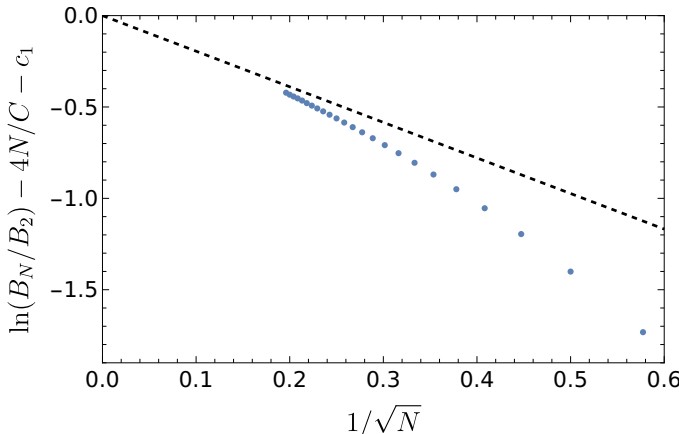

Figure 3: The quantity $\ln(B_N/B_2) - 4N/C - c_1$ as a function of $1/\sqrt{N}$. The data for $B_N/B_2$ are taken from Ref. [17] and the dashed line stands for $-2\sqrt{2}/\sqrt{NM_2}$, see Eq. (28).

which exceeds the Bogoliubov accuracy by half a power of $|g| \sim 1/N$. To check this scenario, in Fig. 3 we show the data of Ref. [17] in the form $\ln(B_N/B_2) - 4N/C - c_1$ (where we use $c_1 = -1.9067$) as a function of $1/\sqrt{N}$ together with the line $-2\sqrt{2}/\sqrt{NM_2}$ (dashed). We see that the presence of the $1/\sqrt{N}$ term is a reasonable hypothesis, also consistent with the fact that $c_1 = -1.91(1)$ found here is rather far from $c_1 \approx -2.06(4)$, obtained in Ref. [17] assuming only integer powers. For firmly proving or disproving this hypothesis one needs to reach higher values of $N$, which is possible with current numerical techniques [22].

In some sense the quantum zero-point motion of the collective coordinate $R$ can be considered as a partial restoration of the mean-field Pitaevskii-Rosch scaling symmetry. Obviously, the other symmetries [translational and $U(1)$] are completely unbroken in the quantum case and the corresponding coordinates should remain completely delocalized. Quantizing the corresponding modes, as we have just done for the breathing mode, we would get vanishing zero-point energies. This is why our symmetry-breaking Bogoliubov method is sufficient for determining the energy at the Bogoliubov level. This statement may still seem surprising since a general number-nonconserving theory should in principle allow for increasing $N$ and, therefore, exponentially descending in energy according to Eq. (24). The reason why the Bogoliubov approach works is that it does not allow significant changes of the wave function (large fluctuations of $R$). To remove any shadow of doubt we have directly checked that the number-conserving approach of Castin and Dum [26] predicts the same soliton energy. It may be useful to provide a few details here.

To zeroth order, the number-conserving theory of Ref. [26] assumes that the system is in the Fock state of $N$ atoms occupying the single-particle orbital $|\Psi_R\rangle$ (normalized to 1). For $g = g_c$ the expectation value of the Hamiltonian (13) in this state equals $1/2R^2$, different from the vanishing mean-field energy in the symmetry-breaking approach. This difference results from the quantum-mechanical behavior of the operator $\hat{\Psi}_{\boldsymbol{\rho}}^\dagger \hat{\Psi}_{\boldsymbol{\rho}}^\dagger \hat{\Psi}_{\boldsymbol{\rho}} \hat{\Psi}_{\boldsymbol{\rho}}$ and from the fact that the expectation value of the interaction is proportional to the number of pairs $N(N-1)/2$ and not to $N^2/2$ as implied by the mean-field energy functional Eq. (7). The difference is at the Bogoliubov level and one still needs to take into account other contributions of this order of magnitude. To this end Castin and Dum construct a quadratic Hamiltonian of type (16), but operating on excitations orthogonal to $|\Psi_R\rangle$. The result is Eq. (18), where the operators $\hat{A}$ and $\hat{B}$ are replaced by $\hat{Q}\hat{A}\hat{Q}$ and $\hat{Q}\hat{B}\hat{Q}$ and the operator $\hat{Q} = 1 - |\Psi_R\rangle \langle \Psi_R|$ projects onto single-particle states orthogonal to $\Psi_R(\rho)$.

Castin and Dum explain that the nonzero part of the Bogoliubov spectrum is the same as in the symmetry-breaking approach. Therefore, the sum over eigenenergies in Eq. (17) is also the same. However, we check that the trace changes as $-\mathrm{Tr}(\hat{Q}\hat{A}\hat{Q})/2 + \mathrm{Tr}\hat{A}/2 = -1/2\mathrm{R}^2$, which means that the two approaches indeed predict the same energy at the Bogoliubov level.

# 6 Summary

In this paper we use the Bogoliubov theory to calculate the leading-order quantum correction to the energy of a two-dimensional soliton. Our calculations improve the accuracy of the previously known exponential scalings [13] by fixing the preexponential factors in Eqs. (23) and (24). We also study the classical breathing dynamics of the soliton and find that the frequency of small-amplitude oscillations scales as $\Omega \sim |B_N|/\sqrt{N}$ at large $N$. The very existence of this mode is a clear manifestation of the quantum anomaly discussed in Ref. [18], but now observable in free space. Quantum-mechanically, the breathing excitations form a ladder of states and we conjecture that their number increases one by one as we add particles to the soliton. We also conjecture that the leading beyond-Bogoliubov relative correction to the soliton energy scales as $1/\sqrt{N}$ and originates from zero-point fluctuations of the breathing mode. Proving these conjectures emerges as a sound project for future studies.

**Acknowledgment** We are grateful to G. Astrakharchik, E. Demler, N. Pavloff, and A. Tononi for valuable discussions.

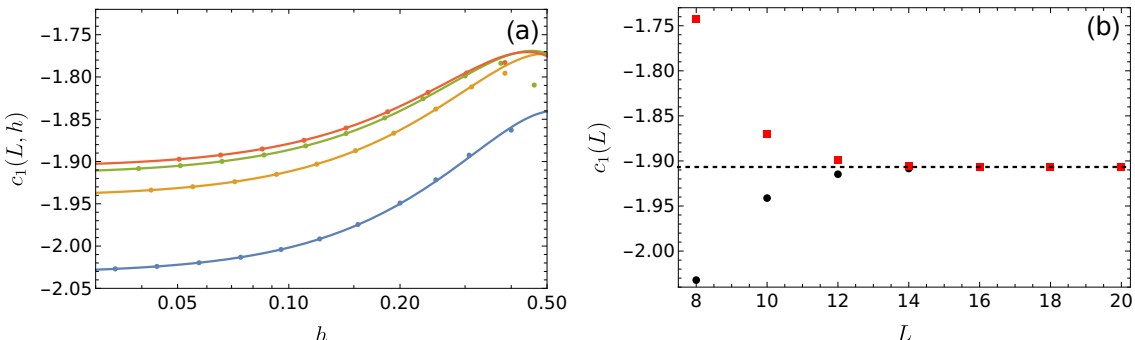

Figure 4: (a) The quantity $c_1(L, h)$ defined in Eq. (29) as a function of $h$ for $L = 8$ (blue), 10 (yellow), 12 (green) and 20 (orange). The curves are fits by the functions $c_1(L) + \alpha(L)h^2 \ln[\beta(L)h]$. (b) The fitting parameter $c_1(L)$ as a function of $L$. The black circles and red squares correspond, respectively, to the Neumann and Dirichlet boundary conditions at the box boundaries. The horizontal dashed line shows $c_1 = -1.9067$.

## A    Determination of $c_1$

Our procedure for calculating $\xi$ and determining the constant $c_1 = -1 - 8 \ln 2 + \ln(C\xi^2)$ in the large-$N$ expansion $\ln(B_N/B_2) = 4N/C + c_1 + ...$ is based on solving Eq. (18) on a square lattice with spacing $h$ and external dimensions $L \times L$. More precisely, instead of Eq. (18) we diagonalize the matrix $(\hat{A} - \hat{B})(\hat{A} + \hat{B})$, eigenvalues of which are $\epsilon_\nu^2$. The Laplacian is defined by Eq. (14) and, without loss of generality, we set $R = 1$. After diagonalization we use Eq. (17) to calculate the quantity

$$c_1(L, h) = -(8/C)E_{\mathrm{BMF}}(L, h) - 1 - 8 \ln 2 + \ln C + 2 \ln h, \qquad (29)$$

which tends to $c_1$ in the limit $h \to 0$, $L \to \infty$.

In our method we place the center of the soliton in the center of the box, which makes the system invariant with respect to reflections $x \to -x$ and $y \to -y$. These symmetries allow us to perform diagonalization only on the quarter of the box, i.e., for $x, y \in [0, L/2]$, although we then have to run the code for three different sets of boundary conditions on the eigenfunctions at the edges $x = 0$ and $y = 0$. In the first case we set the zero-derivative condition on both boundaries (Neumann-Neumann), in the second the zero-function boundary condition on both boundaries (Dirichlet-Dirichlet), and the third case corresponds to Neumann-Dirichlet configuration [the corresponding spectrum should be counted twice in Eq. (17)]. Moreover, to estimate finite size effects we repeat these calculations for two different sets of boundary conditions (Neumann and Dirichlet) at the external boundaries of the box.

In Fig. 4(a) we show $c_1(L, h)$ as a function of $h$ for $L = 8$, 10, 12, and 20 obtained with the Neumann boundary condition at the external boundaries. Results look similarly for other values of $L$ and for the Dirichlet boundary conditions, but we do not show them to avoid clutter. The solid curves are fits assuming $c_1(L, h) = c_1(L) + \alpha(L)h^2 \ln[\beta(L)h]$, where $c_1(L)$, $\alpha(L)$, and $\beta(L)$ are fitting parameters. The fits are based on the four leftmost data points (corresponding to the four smallest $h$), but as one can see, they work very well for significantly larger $h$. As we argue in the main text this form of fitting function is a typical effective-range expansion in a weakly-interacting regime in two dimensions. We have also seen this behavior in the numerical analysis of two-dimensional droplets in Ref. [32].

In Fig. 4(b) we plot the fitting parameter $c_1(L)$ as a function of $L$ for the Neumann (black circles) and for the Dirichlet (red squares) boundary condition at the box edges. Both datasets exponentially converge to $c_1 = -1.9067$ (dashed horizontal line) with the exponent $\propto e^{-3L/4}$ (fits are not shown). We estimate the uncertainty of this final result as the difference between the extrapolated value $c_1(20.)$ and the value $c_1(20., 0.05)$, calculated for the largest $L$ and smallest $h$. We thus claim $c_1 = 1.91(1)$.

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
