# Peer review of "Beyond-mean-field analysis of the Townes soliton and its breathing mode"

_SciPost Physics_

## Round 1 · Referee Report · Anonymous (Referee 1) · 2025-2-8

Strengths

  1. The manuscript highlights subtle issues in the quantum dynamics of cold atomic clouds

  2. The general idea is (relatively) straightforward to follow

  3. The problem is potentially of high relevance in view of next experiments

Weaknesses

  1. There are a few potentially shaky technical points

  2. The presentation is at several points obscure

Report

The manuscript by Petrov reports a theoretical study of the dynamics of an atomic cloud in a quite specific, yet experimentally relevant configuration where quantum effects are expected to play a major role. At the mean-field level, the system displays in fact a scale-independence that allows for zero-cost changes in the cloud size and, thus, a trivial expansion or collapse dynamics. When quantum fluctuation effects are included at the Bogoliubov level, a non-trivial energy landscape appears as a function of the cloud size, which results in breathing oscillations with a peculiar amplitude-depedent frequency. Finally, in order to find a reliable value of the cloud energy (to be compared to microscopic few-body calculations), the zero-point energy of the breathing mode must be included.

In my view, these results are of high interest as they point out a simple configuration where subtle and very non-trivial quantum effects can be observed in the dynamics of a dilute ultracold atomic cloud. The way the macroscopic breathing degree of freedom is singled out from the other higher-frequency modes and receives an energy contribution from these latter is interesting but might be expected on the grounds of previous work (by the same author and by others). On the other hand, the need for a quantized description of the (macroscopic) breathing mode in a sort of "Born-Oppenheimer" picture is much less straightforward and intriguing. The configuration is relatively simple and experimentally accessible to next experiments, so the theoretical study has a high experimental relevance. On this basis, I find that the content of the manuscript is well suitable for being considered for publication on SciPost.

However, I have some concerns on a few technical aspects of the work. These must be duly clarified before I can make a firm recommendation about publication. At the same time, several points of the presentation also need improving to guarantee easy readability of the manuscript. A list of my main concerns follows. I urge the author to take all these remarks in due consideration before resubmitting to SciPost or to any other journal.

1- I am not able to understand what is the role of the parameter \xi that appears in (19) and then in many following equations. Is it a lattice-dependent quantity (as I would guess from the text at the end of Sec.3.1) ? Then, I would infer that the energy landscape in (21) is also lattice-dependent. But then I don't understand why the xi-dependence disappears in the frequency (25), probably hidden in R_N. I am totally lost.... how should one choose \xi to get results to be compared to experiments? Appendix A is too obscure to be really helpful in this sense.

2- In (18), I suspect that g is a sort of 'bare' interaction constant that (for each value of the lattice spacing h) has to be renormalized in order to give the correct energy of the two-body bound state. Is it really so? Then, I am lost about the meaning of g_c... does it has to be renormalized as well?

3- The quantity B_N is used with different meanings in different parts of the manuscript. For instance, before eq.(26) it looks like it is used to indicate the numerically calculated exact ground state energy. In (18) it looks like B_2 is the exact 2-body ground state energy on the lattice, in (23) B_N is the ground state energy within Bogoliubov approximation. I recommend the author to improve the notation to avoid confusion between these distinct quantities.

4- A general question: the theory includes the quantum fluctuations of Bogoliubov theory in the energy landscape but says nothing about the 'kinetic' term. Why the Bogoliubov theory does not provide any correction to the mean-field kinetic energy appearing as the first term of (11), e.g. as a renormalization of the 'mass' ?

5- A few lines above eq.(20), I don't understand why the expression (19) for the energy is only valid next to g=g_c. What would happen away from this point?

6- after (21), the author speaks of a 'effective-range term' in (18). What is he referring to?

7- The adiabaticity condition (20) should be justified a bit better. I would have expected it should involve degrees of freedom other than the one related to R, but this does not appear to be the case of the formula. The author should check that this is really the required condition.

8- The discussion around (26) about the need of including the zero-point energy associated to the dynamics along R is one of the most exciting points of this manuscript. Unfortunately, however, it is discussed in a quite obscure way. I recommend the author to work on improving its presentation.

9- Right after eq.(15), the authors speaks of four pairs of eigenstates with \epsilon=0. Is he sure about this? I vaguely remember that in the presence of broken symmetries the Bogolibov operator is non-diagonalizable (unless some special care is paid to conserved quantities, e.g. in particle-number-conserving Bogoliubov theories). I suspect that some feature of this kind is also present here. The authors should clarify.

10- Right after eq.(8), the author speaks of 'minimizing' the action with respect to \theta to remove this variable from the Lagrangian. I suspect this is legitimate as the Lagrangian does not contain terms involving the time-derivative of \theta. Is this correct? Perhaps it is worth mentioning it and justifying it as it is not so straightforward.

11- What is \mu^\prime in eq.(11) ?

12- The in-line integral right before eq.(9) looks suspicious. Is there a typo?

13- A few lines from the bottom of pag.7, the author should define precisely what R_rms is. How is the average of the root mean square taken? Is it an average over time along the orbit?

14- As a general remark, it would be useful if the author could connect his work to other related theories (in cold atoms, but also in molecular physics and other fields is possible) that adiabatically eliminate microscopic degrees of freedom to write effective quantum Hamiltonians for the macroscopic degrees of freedom.

Requested changes

  1. Clarify all the technical issues

  2. Make an effort to improve the presentation

  3. Take into account my remarks to remove typos and ambiguities

Recommendation

Ask for minor revision

---

## Round 1 · Referee Report · Anonymous (Referee 2) · 2025-3-10

Strengths

  1. The use of the advanced method to expand above the mean-field manifold is well-justified and adds significant depth to the analysis.

  2. The manuscript is well-organized, with a clear introduction, detailed methodology, and a comprehensive discussion of results.

Weaknesses

  1. The manuscript is highly technical at some points. While this is expected in some theoretical works, a more detailed explanation of some parts could make the work more accessible to a larger audience.

  2. The manuscript at certain points appeared rushed with some statements or results directly written with no clear explanation.

Report

The manuscript presents a theoretical analysis of the Townes soliton and its breathing mode in two-dimensional bosonic systems with zero-range attractive interactions. Using Bogoliubov perturbation theory, the authors go beyond the mean-field description, providing improved results for the ground-state energy B_N and size R_N., and the scaling of frequency \Omega of the breathing mode.
However, I have certain points that need to be addressing in my opinion:

1. It would be helpful if the authors show few steps or cite references to show how one can obtain Eq. 13 from Eq. (12) using the wavefunction of standard Bogoliubov theory.

2. What is \mu’ is Eq. (10).

3. In Eq 14 what is the cut-off (if any) or values of $\nu$ and how do you decide that?

4. “A peculiarity of the Bogoliubov spectrum in our case is that Eq. (14) has four pairs of eigenstates with 𝜖=0.” Is there a way this can be presented or shown more transparently since it doesn’t appear straightforward to me why these should be zero here? The authors do detail on the 4 modes this is related to but if possible then please add a line or two for some kind of further explanation in this regard (if possible).

5. It is not clear how in Eq 16 the first line is equal to the second one. Please add more details for the same.

6. “Although the lattice model gives a way to calculate observables in terms of g and h, the final results will be expressed in terms of the unique parameter B_2”, could authors please write one line verifying that if B_2 is Dimer binding energy?

7. “In fact, by rescaling the coordinate in Eq. (14) one can show that the beyond-mean-field correction is a product of 1/R^2 and a function of h/R, which, as we have just argued, is logarithmic when h/R << 1” Is there a way to explicitly show that? If yes, then I would be interested to see a few steps for details on this statement.

8. If I simply look at the derivative of Eq 21 to obtain R_N (minimum value) then there is a slight discrepancy with the constant mainly 4\sqrt(2) which could be absorbed in \xi since it is some constant. However, since not much is known about \xi clearly., also, since I could be making mistakes here, so I request authors to check and verify if someone tries to just find minima by differentiation Eq. 21, can Eq. 22 follow from there simply or there’s some more information needed to verify this result?

Requested changes

  1. Please address the above concerns
  2. If possible please add some more information or citations at certain technical points that could help authors reach a wider audience.

Recommendation

Ask for minor revision

---

## Round 2 · Referee Report · Anonymous (Referee 1) · 2025-5-18

Strengths

  1. The manuscript highlights subtle issues in the quantum dynamics of cold atomic clouds

  2. The general idea is (relatively) straightforward to follow

  3. The problem is potentially of high relevance in view of next experiments

Weaknesses

  1. No longer applicable. The main weaknesses have been solved

Report

The changes done on the manuscript by the author are satisfactory. I can now recommend publication on SciPost Physics.

A couple of minor optional (but warmly welcome) further suggestions:

-the abstract is still a bit technical. E.g., I don't like having formulas in there. The author may consider reformulating it in a more accessible way. Also the last paragraph of the intro may be made more accessible and slightly more detailed.

-on pag.5, I am puzzled by the formula |g-gc|~g^2<<|gc|. I don't see what it means to formally set g=gc. The author should spend a few words to explain what he has in mind.

-when discussing after eq.(27) the condition for adiabaticity, the author mentions the risk of non-adiabatic features at large R, but says nothing on what may happen at smnall R. Even though the gap is here the largest, I see from Fig.1 that the effective potential grows very fast. I am therefore wondering if the strong acceleration felt around this left turning point may induce any additional adiabatic effect.

Requested changes

  1. Take into due consideration my suggestions for (optional) revisions

Recommendation

Publish (surpasses expectations and criteria for this Journal; among top 10%)

  • validity: high
  • significance: top
  • originality: top
  • clarity: high
  • formatting: perfect
  • grammar: perfect

Author:  Dmitry Petrov  on 2025-05-27  [id 5529]

(in reply to Report 1 on 2025-05-18)
Category:
answer to question

I warmly thank Referee #1 for this report and suggestions. Here is my response.

Referee:
-the abstract is still a bit technical. E.g., I don't like having formulas in there. The author may consider reformulating it in a more accessible way. Also the last paragraph of the intro may be made more accessible and slightly more detailed.

Response:
This formula is there on purpose to attract attention. In any case, given its level of complexity, I think that this is really a question of personal style. I take this opportunity to make the following change in the abstract, hopefully compatible with the Referee's request as it increases the text-to-formula ratio and makes the abstract slightly more informative. I replace the sentence

"The oscillatory breathing motion results from the quantum-mechanical breaking of the mean-field scaling symmetry."

by

"The breathing oscillations, absent on the mean-field level, result from the quantum-mechanical breaking of the mean-field scale symmetry."

In the last paragraph of the introduction I added a couple of clarifying sentences about the preexponential term and about conjectures made at the end of the article.

Referee:
-on pag.5, I am puzzled by the formula |g-gc|~g^2<<|gc|. I don't see what it means to formally set g=gc. The author should spend a few words to explain what he has in mind.

Response:
I am interested in the systematic expansion of the energy in powers of g. The Bogoliubov level corresponds to terms ~g^2. The beyond-mean-field correction is of this order of magnitude. Thus, when calculating it with the Bogoliubov accuracy I can replace g by g_c since the difference g-g_c is of higher order. I think this should be understandable from the context (see the paragraph preceding the discussion and the last paragraph of Sec.2). Nevertheless, I added the sentence "Distinguishing between $g$ and $g_c$ in this case would exceed the Bogoliubov accuracy." To avoid confusion I also replaced "to formally set g=g_c" by "to use the approximation g=g_c". I hope this is more clear now.

All this discussion would be much more straightforward, if I could just set g=g_c from the very beginning. However, as I explain in the text, I cannot do this because of the constraints on the potential range. So, I have to keep g different from g_c, but the difference can be chosen to be of higher order. This choice is made possible by the fact that the dependence of g on h is very slow and that effective-range corrections are algebraic in h, i.e., exponentially small as a function of g.

Referee:
-when discussing after eq.(27) the condition for adiabaticity, the author mentions the risk of non-adiabatic features at large R, but says nothing on what may happen at smnall R. Even though the gap is here the largest, I see from Fig.1 that the effective potential grows very fast. I am therefore wondering if the strong acceleration felt around this left turning point may induce any additional adiabatic effect.

Response:
As far as I know, the adiabatic theorem does not impose constraints on the acceleration.

I guess that account of the acceleration may lead to quantitative corrections, but I do not know how important they are. In the paper I argue that breaking of the adiabatic condition at large R may lead to a qualitatively different description of the higher part of the spectrum just below the continuum. It is not obvious to me that nonadiabatic effects at the left turning point can lead to qualitatively important consequences. At least, being close or far from the threshold E=0 does not seem to play a big role there. We can as well discuss the left and right turning points of small-amplitude oscillations.

---

## Round 2 · Referee Report · Anonymous (Referee 2) · 2025-5-23

Strengths

  1. The manuscript now adds more details and insights on calculations.

  2. Presents advanced analysis to expand previous results beyong mean-field manifold.

Weaknesses

Nothing in particular since all the comments have been taken care of.

Report

The authors have meticulously added all the details and comments referees requested, and now the manuscript appears clearer.

Requested changes

None.

Recommendation

Publish (surpasses expectations and criteria for this Journal; among top 10%)

  • validity: high
  • significance: high
  • originality: high
  • clarity: high
  • formatting: good
  • grammar: good

Author:  Dmitry Petrov  on 2025-05-28  [id 5530]

(in reply to Report 2 on 2025-05-23)

I warmly thank the referee.

---

## Round 2 · Author Response

Dear Editors,

I am grateful to the Referees for their careful reading of the manuscript, for their very positive opinion, and for valuable comments and suggestions. I have modified the manuscript according to their suggestions. I hope that you find the revised version suitable for publication.

Sincerely,
Dmitry Petrov

---

## Round 2 · List of Changes

The derivation of the mean-field action is rewritten and I discuss the mean-field evolution in more detail (Sec.2).
The hierarchy of energy scales and the logic behind the perturbative expansion are explained in a new paragraph in the end of Sec.2.
I modified the derivation of the formula for the beyond-mean-field correction (Sec.3). The regularization is now introduced from the very beginning, the logarithmic dependence on the cutoff is derived explicitly, and the meaning of xi is explained more clearly.
I modified the discussion on the adiabaticity in Sec.4. I added a few references as recommended by Referee 1.
A few misprints are fixed.
No modification of the results or figures.

---

## Round 3 · Author Response

I thank the referees for the reports. I slightly modify the manuscript in response to Referee #1. I hope you find it suitable for publication.
Sincerely,
Dmitry Petrov

---

## Round 3 · List of Changes

1) In the abstract I replaced the sentence
"The oscillatory breathing motion results from the quantum-mechanical breaking of the mean-field scaling symmetry."
by
"The breathing oscillations, absent on the mean-field level, result from the quantum-mechanical breaking of the mean-field scale symmetry."
2) In the last paragraph of the introduction I added two sentences:
"We thus predict the preexponential factor in the dependence of the ground state energy on the particle number."
and
"We also make a few conjectures concerning the next-order correction to the soliton's energy and the nature of its excitation spectrum."
3) After Eq.(12) in the paragraph starting with "The equation of motion corresponding..." I replaced R(t)^2 by R^2(t).
4) On page 5 in the paragraph starting with "A technical difficulty with..." I replaced the phrase "to formally set g = g_c" by "to use the approximation g = g_c" and added the sentence
"Distinguishing between g and g_c in this case would exceed the Bogoliubov accuracy."

---

## Editorial Decision

editorial_decision: